# Improving the Accuracy of the Effective Atomic Number (EAN) and Relative Electron Density (RED) with Stoichiometric Calibration on PCD-CT Images

**DOI:** 10.3390/s22239220

**Published:** 2022-11-27

**Authors:** Kihong Son, Daehong Kim, Sooyeul Lee

**Affiliations:** 1Medical Information Research Section, Electronics and Telecommunications Research Institute, Daejeon 34129, Republic of Korea; 2Department of Radiological Science, Eulji University, Seongnam 13135, Republic of Korea

**Keywords:** PCD-CT, dual-energy, EAN, RED, stoichiometric calibration

## Abstract

The photon counting detector (PCD) in computed tomography (CT) can count the number of incoming photons in order to obtain energy information for photons corresponding to user-defined thresholds. Research on the extraction of effective atomic number (EAN) and relative electron density (RED) using dual-energy CT (DECT) is currently underway. This study proposes a method for improving EAN and RED accuracy of tissue-equivalent materials by using PCD-CT-based stoichiometric calibration. After obtaining DECT images in energy bin (EB) and full spectrum (FS) modes for eight tissue-equivalent materials, the EAN was calculated with stoichiometric calibration. Using the EAN image, the RED image was acquired to evaluate the accuracy. The errors of both EAN and RED obtained with EB were within 4%. In particular, the accuracy of RED was higher than that of the FS method. Study results indicate that PCD-CT contributes to improving EAN and RED accuracy. Further studies will be aimed at reducing ring artifacts by pixel-correcting PCD images and improving stopping power ratio (SPR) measurements for dose calculation in particle therapy.

## 1. Introduction

Dual-energy CT (DECT) is a useful tool for diagnosing diseases in clinical practice, and can identify human tissue characteristics based on the fact that the two photon spectra of different energy levels differ in the degree of attenuation caused by a particular material [1]. Due to the difference in attenuation coefficients, DECT is applied in various ways to imaging, including material separation, tissue characterization, vascular imaging, lesion imaging, and artifact reduction [2,3,4,5,6]. In addition, DECT can be used in the field of radiotherapy, particularly, it can improve dose accuracy in radiotherapy plans. Radiation therapy plans are based on the relative electron density (RED) of the CT image to determine the distribution of patient dose. The RED exists when the relative electron density is fitted to the Hounsfield unit (HU) of the human equivalent material, as measured by single-energy CT (SECT). In contrast, DECT provides more accurate relative electron density information than conventional SECT techniques since it calculates relative electron density values based on the effective atomic number of human equivalents, thereby enhancing the accuracy of dose distribution in patients [7].

Previously, energy-integrating detector (EID)-based CT was primarily used to acquire dual-energy images. However, the EID is insensitive to spectral information because it accumulates an electric charge at each detector pixel caused by photon interaction during the acquisition process regardless of the energy of the photons. The detection mechanism does not provide sufficient contrast between substances and does not reflect specific information from tissues [8]. PCDs are being used to solve this problem in X-ray imaging. In contrast to EIDs, PCDs with energy discrimination are capable of counting the photons incident on the detector. Further, the PCD provides energy information in multicolored X-ray spectra using multienergy thresholds, so it can be applied to new X-ray imaging methods such as DECT [9].

The accuracy of the effective atomic number (EAN) and RED of materials has been improved by analyzing DECT images. Goodsitt et al. [10] reported that the differences between the experimental and theoretical results based on EID CT for liver and brain were −6% and 15%, respectively. Additionally, Tatsugami et al. [11] showed that the EAN of adipose tissue and bone tissue was inaccurate by about −8% and 7%, respectively. The use of PCD-CT for EAN and RED imaging was also investigated. Calculations of stopping power ratios (SPRs), non-destructive testing, and EAN and RED measurements were performed for elements and compounds. From the results of Lee et al. [12], the maximum error of EAN and RED obtained using the proposed new fitting method was 20% and −15.7% in graphite material. Zou et al. [13] found the maximum errors for EAN and RED in Ti were 16.8% and 40%, respectively. Studies related to non-destructive testing have revealed a maximum error of EAN of 4.12% in Al and a maximum error of RED of −2.99% in PTFE [14]. According to Yamashita et al. [15], EAN imaging showed better results compared with DECT and mono X-ray measurements. EAN and RED values are critical not only for their role as diagnostic images for disease detection, but also for the accuracy of radiation therapy dose calculations. To increase the accuracy of EAN and RED, previous studies have used software and hardware methods, such as the development of fitting methods and the use of PCD-CT. Nevertheless, there were large errors between the theoretical and experimental values for specific materials appearing in PCD-CT-based images, and it is necessary to elucidate the accuracy of EAN and RED for various human equivalents. Therefore, this study aims to improve the accuracy of EAN and RED based on dual-energy PCD-CT measurements of various tissue-equivalent materials. The ratio of HU values of low and high energy to EAN was estimated with stoichiometric calibration.

## 2. Materials and Methods

### 2.1. Stoichiometric Calibration

#### 2.1.1. Extraction of EAN and RED

Jung et al. reported that the stoichiometric calibration can enhance the accuracy of EAN [16]. Further, Bourque et al. demonstrated that stoichiometric calibration can improve the accuracy of RED without requiring spectral measurements or linear hardening corrections [17]. In this study, the EAN and RED were extracted using stoichiometric calibration.

The method for stoichiometric calibration was first introduced by Schneider [18]. Equation (1) can be used to calculate the X-ray linear attenuation coefficient of a material.
(1)$$\mu =\rho \frac{N_{A}}{A}\left[ZK^{KN}\left(E\right)+Z^{n}K^{SCA}\left(E\right)+Z^{m}K^{PE}\left(E\right)\right],$$
where *ρ*, *N_A_*, *A*, *Z*, *K^PE^*, *K^SCA^*, and *K^KN^* are the mass density, Avogadro’s number, atomic weight, atomic number, photoelectric effect, Rayleigh scattering, and Compton scattering. According to Rutherford et al., the values *m* and *n* assigned to energy and substances present in human tissue in kV X-ray images were 4.62 and 2.86, respectively.

It is important to take into account the ratio of the attenuation coefficient of different substances to the attenuation coefficient of water at the same energy when determining the CT number. Therefore, Equation (1) can be transformed into Equation (2) in a similar manner.
(2)$$\frac{\mu }{\mu_{w}}=\frac{\rho \frac{N_{A}}{A}ZK^{KN}\left(E\right)\left[1+Z^{1.86}k_{1}\left(E\right)+Z^{3.62}k_{2}\left(E\right)\right]}{\rho_{w}\frac{N_{A}}{A_{w}}Z_{w}K^{KN}\left(E\right)\left[1+Z_{w}^{1.86}k_{1}\left(E\right)+Z_{w}^{3.62}k_{2}\left(E\right)\right]},$$

According to Equation (2), *ρ_w_*, *Z_w_*, and *A_w_* represent the density, effective atomic number, and atomic weight of water, respectively. *k*_1_ is expressed as $\frac{K^{SCA}}{K^{KN}}$ and *k*_2_ is $\frac{K^{PE}}{K^{KN}}$. In addition, the RED ${\hat{\rho }}_{e}$ is $\frac{\rho \frac{N_{A}}{A}Z}{\rho_{w}\frac{N_{A}}{A_{w}}Z_{w}}$. Equation (3) can be used to express the *HU* as a unit of the CT number.
(3)$$HU=\left(\frac{\mu }{\mu_{w}}-1\right)\times 1000,$$

Equation (3) can be rewritten using Equation (2) as Equation (4).
(4)$$\frac{HU}{1000}+1={\hat{\rho }}_{e}\frac{\left[1+Z^{1.86}k_{1}\left(E\right)+Z^{3.62}k_{2}\left(E\right)\right]}{\left[1+Z_{w}^{1.86}k_{1}\left(E\right)+Z_{w}^{3.62}k_{2}\left(E\right)\right]},$$

We determine the ratio between the two HU values for low and high energy for EAN. Therefore, Equation (4) can be expanded into an expression for the ratio of two energies as Equation (5).
(5)$$\frac{HU_{L}+1000}{HU_{H}+1000}=\frac{1+Z^{1.86}k_{1,L}+Z^{3.62}k_{2,L}}{1+Z_{w}^{1.86}k_{1,L}+Z_{w}^{3.62}k_{2,L}}\times \frac{1+Z_{w}^{1.86}k_{1,H}+Z_{w}^{3.62}k_{2,H}}{1+Z^{1.86}k_{1,H}+Z^{3.62}k_{2,H}},$$

*Z* here refers to the effective atomic number of a substance. *K*_1,*i*_, *K*_2,*i*_(*I* = *L*, *H*) are the spectral parameters for low energy and high energy, respectively. After obtaining the spectral parameter *k*, the EANs of various human equivalents were extracted from the ratio of the two HUs. Equation (5) was solved using a curve fitting function in Matlab (R2013, MathWorks Inc., Natick, MA, USA). Equation (4) may be used to obtain the relative electron density by substituting the effective atomic number derived from Equation (5) into the equation. Equation (4) can be reorganized into an equation for RED as shown in Equation (6).
(6)$${\hat{\rho }}_{e}=\left(\frac{HU}{1000}+1\right)/\frac{\left[1+Z^{1.86}k_{1}\left(E\right)+Z^{3.62}k_{2}\left(E\right)\right]}{\left[1+Z_{w}^{1.86}k_{1}\left(E\right)+Z_{w}^{3.62}k_{2}\left(E\right)\right]},$$

In this case, if a high-energy image is used for the HU, *k*_1_(*E*) and *k*_2_(*E*) will also use high energy values.

#### 2.1.2. Tissue-Equivalent Materials

Stoichiometric calibration was conducted using materials with various effective atomic numbers. The tissue-equivalent rods were inserted into a phantom (Multi-Energy CT Phantom, Sun Nuclear Co., Middleton, WI, USA). Figure 1 illustrates the phantom used in this study, and Table 1 shows the effective atomic number and relative electron density of the various phantom materials. The rod located in the phantom number in Table 1 is located in the same location as in the phantom number in Figure 1. Adipose has the lowest atomic number, with an effective atomic number of 6.44, and cortical bone has the highest effective atomic number of 13.29. The majority of human tissue has an effective atomic number between adipose and cortical bone.

### 2.2. PCD-CT System

#### 2.2.1. Specification of PCD-CT

Figure 2 illustrates the components of a PCD-CT, which include a detector, a source, and a rotating platform. In terms of detectors, PCDs (XC-TDI, Varex imaging, UT) are made of cadmium telluride (CdTe) with pixels of 100 μm in size and 100 μm in pitch. It has a matrix of 3584 × 60 pixels, and the active area is 360 × 6 mm^2^, arranged as a line detector. This detector operates only in the photon counting mode, and the X-ray energy spectrum can be divided into two bins. The X-ray source (RAD-14, Varian Medical Systems, UT) was installed perpendicular to the detector surface. A tube voltage of up to 150 kV can be applied to the source, which is a rotary anode type. A rotating platform is located between the source and the detector. A subject may be placed on the platform, and may rotate while a source and detector are fixed in order to obtain a CT image. The source-to-object distance (SOD) is 829.0 mm and the object-to-detector distance (ODD) from the detector is 508.4 mm.

#### 2.2.2. Data Acquisition

Figure 1 shows a phantom that was scanned with dual energy and reconstructed using the Feldkamp–Davis–Kress (FDK) algorithm and Hann filter. A photon counting method was used to acquire the image. We obtained two data sets in order to confirm the energy bin effect of the photon coefficient detector. First, photons were measured in two vacant regions within one spectrum. Then, 125 kV X-ray spectrum energy bins (EBs) were divided into low-energy (20–60 keV) and high-energy (60–125 keV) regions in order to ensure that the number of photons was similar. A value of 20 keV or less was not used due to the increased noise. The second method involved scanning twice with FS mode to obtain a dual-energy image. Low- and high-energy outputs were powered by 60 kV and 125 kV voltages, respectively. Figure 3 shows images scanned in EB and FS modes. A linear attenuation coefficient was used to express the pixel value of the image. The EAN and RED images obtained by the EB and FS methods were extracted using Equations (5) and (6).

## 3. Results

### 3.1. Stoichiometric Calibration

Figure 3 illustrates how the dual-energy CT images obtained by the EB and FS methods were converted to EAN and RED images using Equations (5) and (6). Equation (5) was used to perform a stoichiometric calibration using a dual-energy image scanned in EB mode. Using Equation (5), the reduction ratio was calculated by dividing the low-energy image by the high-energy image based on the pixel values of the low (EB) and high (EB) images. After obtaining the reduction ratios of the eight substances, the effective atomic number corresponding to the reduction ratio was matched, and curve fitting was performed. Figure 4 shows curve fitting using Equation (5). Figure 4a illustrates the curve obtained from a dual-energy image scanning in EB mode. The horizontal axis represents the EAN, while the vertical axis represents the reduction ratio. The solid red line in Figure 4 indicates the fitting values based on Equation (5), in which R^2^ was 0.9970. By using the fitting procedure, the spectral parameters k_1,L_, k_2,L_, k_1,H_, and k_2,H_ in Equation (5) were determined. Table 2 presents the spectral parameters obtained. The stoichiometric calibration of dual-energy photography was performed in the same manner as for BS. The R^2^ value was 0.9966. In Figure 4b, an image obtained in FS mode is used to demonstrate the reduction ratio for EAN. The spectral parameters obtained in FS mode are also presented in Table 2.

The REDs of EB and FS were calculated using the eight substances in Table 1 following Equation (6). The horizontal axis was set to the EAN of the material and the vertical axis to the RED, and then fitted to Equation (6). In Equation (6), one of the low-energy and high-energy images is used in the left side HU. Table 3 describes the parameters k_1_(E) and k_2_(E) obtained through this method. The R^2^ values in EB and FS were 0.9722 and 0.9636, respectively.

### 3.2. EAN and RED Results

Figure 5 shows the tissue-equivalent materials of the EAN and RED images. The upper line in Figure 5 represents EAN and RED images acquired in EB mode. The lower line in Figure 5 indicates EAN and RED obtained in FS mode. In Figure 6, the theoretical and experimental values of EAN and RED are illustrated in a graph.

Figure 6a shows EAN obtained in the EB and FS modes. In EB mode, comparing the measured EAN with the theoretical value, the deviation was between −3.42% and 3.95%. The average error was 2.48% and the standard deviation was 1.23%. In FS mode, the measured EAN deviated from the range of −3.49% to 2.93% compared to the theoretical value. The average error was 2.04% and the standard deviation was 1.03%. As shown in Figure 6b, RED was calculated using the EB and FS methods. In EB mode, as compared to the theoretical value, the measured RED varied between −0.65% and 3.57%. The average error was 1.68% and the standard deviation was 1.35%. In FS mode, the measured RED varied from −1.64% to 9.55% compared to the theoretical value. The average error was 4.58% and the standard deviation was 3.26%.

Table 4 summarizes the theoretical and experimental values of EAN and RED obtained in EB and FS modes. Parentheses indicate the error of the experimental value compared to the theoretical value. Based on EAN, the maximum errors were 3.95% in EB mode and −3.49% in FS mode. The overall average error in EB and FS was 2.48% and 2.04%, respectively, which was 0.44% lower in FS than in EB. Regarding RED values, the maximum errors in EB and FS modes were 3.57% and 9.55%, respectively. The overall average error was 1.68% and 4.58% for EB and FS modes, respectively.

## 4. Discussion

The purpose of this study was to increase the accuracy of EAN and RED of various tissue-equivalent materials based on dual-energy PCD-CT. A stoichiometric calibration method was used to increase the accuracy of determining the effective atomic number of the human equivalents. EAN and RED were extracted by dual-energy CT using EB and FS in order to compare their accuracy. A photon counting mode was used to acquire both the EB and FS images.

Based on previous studies, the average error of EAN extracted with EB and FS was 2.48% and 2.04%, respectively, and it was confirmed that FS had a slightly higher accuracy than EB. According to the stoichiometric calibration used for curve fitting, EB had an R^2^ of 0.9970, and FS had an R^2^ of 0.9966. The ring artifact, however, was notable in the EAN image of EB, therefore the measurement error is considered to be high. The RED extracted with EB showed a better result than the FS result with an average absolute error of 1.68%. Particularly, RED had improved accuracy in adipose and breast tissue, whereas EAN had a lower accuracy. The RED image did not display more ring artifacts than the EAN image. EB also produced a better result in curve fitting than FS, resulting in an R^2^ of 0.9722 versus 0.9636 for FS.

The EAN of various tissue-equivalent materials was calculated as 2.94 in the study. Schaeffer et al. [19] reported that it is most appropriate for EAN calculations because it produces the smallest matching interval in the head and body phantom when the EAN value is set as 2.94. The diameter of the phantom used in this study is 20 cm, which is close to the circumference of the head. The reason why the image of the body phantom was not obtained is that the FOV is unable to scan the size of the body phantom on the geometry of our PCD-CT.

As part of the stoichiometric calibration, eight substances were used: adipose, breast, solid water, brain, liver, iodine (2 mg/mL), inner bone, and cortical bone. Calibration curves are determined by the attenuation ratio between low and high energies of human organs. Due to the low X-ray attenuation of the lung, it has a relatively large percentage deviation among the major organs of the human body. The heterogeneity of the lung equivalent material and the low electron density value contribute to large variations in electron density values [20]. For this reason, calibration was not performed for lung in this study.

In previous studies, the effective atomic number range selected for the human equivalents was 6 to 15. The range of EAN in this study fell within that range, from adipose tissue (Z_eff_ = 6.44) to cortical bone (Z_eff_ = 13.29). EANs and REDs extracted from tissue-equivalent materials have primarily been used to improve the classification of materials and increase the accuracy of dose calculations in radiation treatment planning. An EID was the CT detector used in these previous studies. The X-ray energy detected by the EID CT is proportional to the detector signal. Since X-rays are primarily detected by charge accumulation, they do not have an energy decomposition function. However, PCD-CT calculates each individual photon equally regardless of its energy measurement. Therefore, a photon may be detected based on an energy threshold defined by the user. By utilizing the advantages of PCD-CT, attenuation information of specific materials can be accurately obtained [21].

The advantage of energy discriminating from PCD-CT has led to many studies on the extraction of EAN and RED. In addition, it has been used to improve the accuracy of dose calculations in particle treatments such as protons [12]. PCD-CT has also been used to reduce the artifact caused by the beam-hardening effect in the field of material science [14]. However, previous studies utilizing PCD-CT have shown that both EAN and RED results are high in error, and no improvement is discernable for EAN and RED for tissue-equivalent materials. In this study, EAN and RED were extracted from tissue-equivalent materials with a range from 6 to 15 in EAN. Higher accuracy in material separation was obtained compared to previous studies. Particularly, the maximum errors for EAN and RED extracted in EB mode were within 4% and 4%, respectively.

Stopping power ratio (SPR) is crucial to accurately calculate a dose in particle therapy as the demand for particle therapy has increased in recent years. Unlike SECT, DECT is of interest as an imaging modality for proton therapy planning because it can discriminate changes in patient density and chemical composition. In the proton treatment, the SPR of a given medium to water is used to calculate the water-equivalent path length (WEPL) of the patient’s tissue. WEPL is also used to calculate initial proton energies and adjust water equivalent dose to tissue dose. Therefore, the accuracy of SPR is important in contributing to the dose accuracy of protons. In order to improve the accuracy of SPR calculated from EAN and RED, a PCD-based detection would be used to study the accuracy improvement of EAN and RED. Once the accuracy of SPR measurements can be ensured by using PCD-CT, dose calculation will be more accurate in terms of patient-specific dose information, compared to the SPR obtained from the traditional SECT.

The pixel matrix of the PCD used in this study was 5139 × 60. The ring artifact was evident in the EAN image extracted from the EB mode, showing slightly lower accuracy even though the R^2^ value was higher than FS. Therefore, future research requires efforts to increase PCD image quality by developing a method for correcting ring artifacts.

## 5. Conclusions

A method for improving the accuracy of EAN and RED has been proposed using both PCD-CT-based dual-energy images and stoichiometric calibration. The EB method was used to improve the accuracy of EAN and RED. This result indicates that PCD-CT will be helpful in accurate classification of substance information and in ensuring the accuracy of dose distribution. It is necessary to perform pixel correction in future studies in order to improve the EAN image quality of PCD-CT. SPR extraction studies that can assist in improving the dose accuracy in particle therapy are also required.

## Figures and Tables

**Figure 1 sensors-22-09220-f001:**
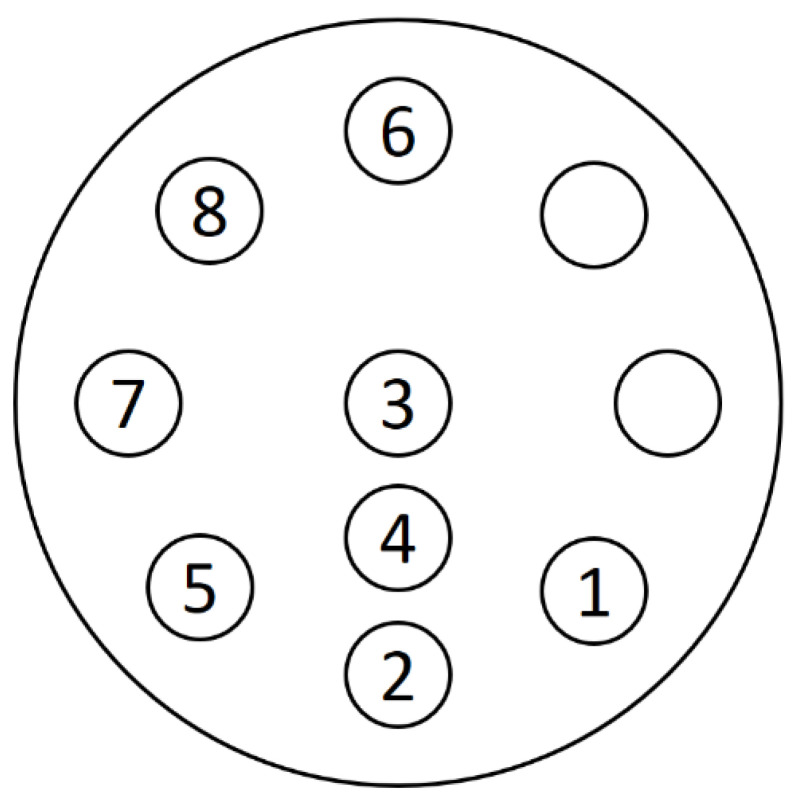
A phantom containing eight tissue-equivalent materials. The numbers are the same as the substances in Table 1.

**Figure 2 sensors-22-09220-f002:**
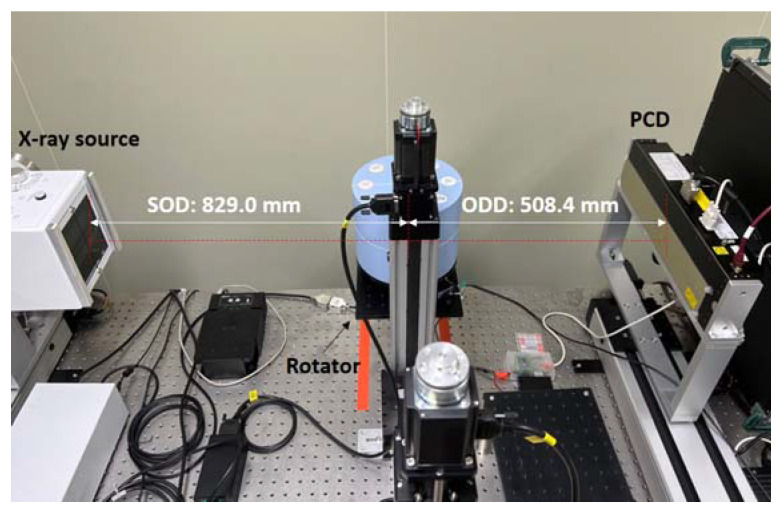
A photon counting detector (PCD) computed tomography (CT) system used in this study. There is a rotating platform between the X-ray source and the line type detector. Abbreviations: source to object distance (SOD); object to detector distance (ODD).

**Figure 3 sensors-22-09220-f003:**
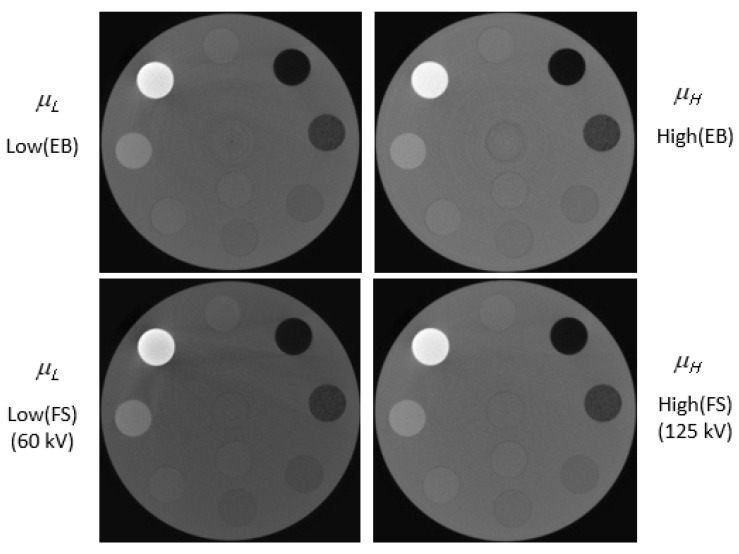
Dual-energy CT images shown with linear attenuation coefficients acquired in energy bin (EB) and full spectrum (FS) modes.

**Figure 4 sensors-22-09220-f004:**
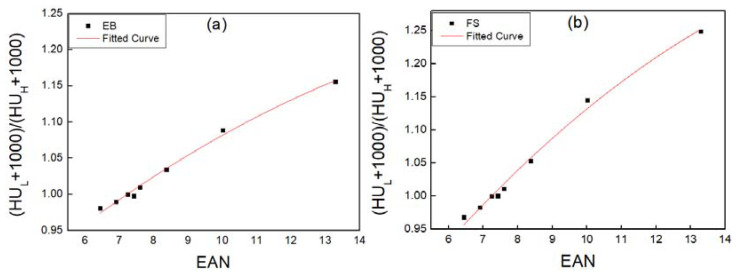
Stoichiometric curve fitting results as a function of effective atomic number (EAN) and the linear attenuation ratio for (**a**) energy bin (EB) and (**b**) full spectrum (FS) mode.

**Figure 5 sensors-22-09220-f005:**
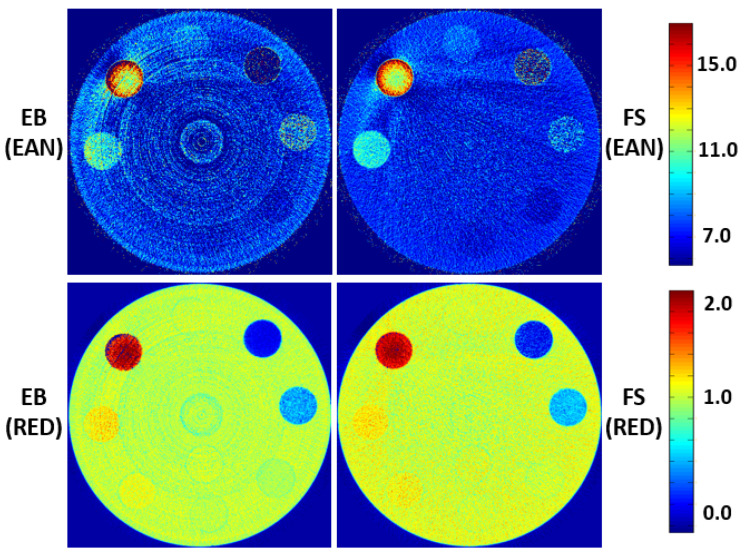
The upper row indicates effective atomic number (EAN) and relative electron density (RED) images acquired in energy bin (EB) mode. Bottom row is EAN and RED images acquired in full spectrum (FS) mode.

**Figure 6 sensors-22-09220-f006:**
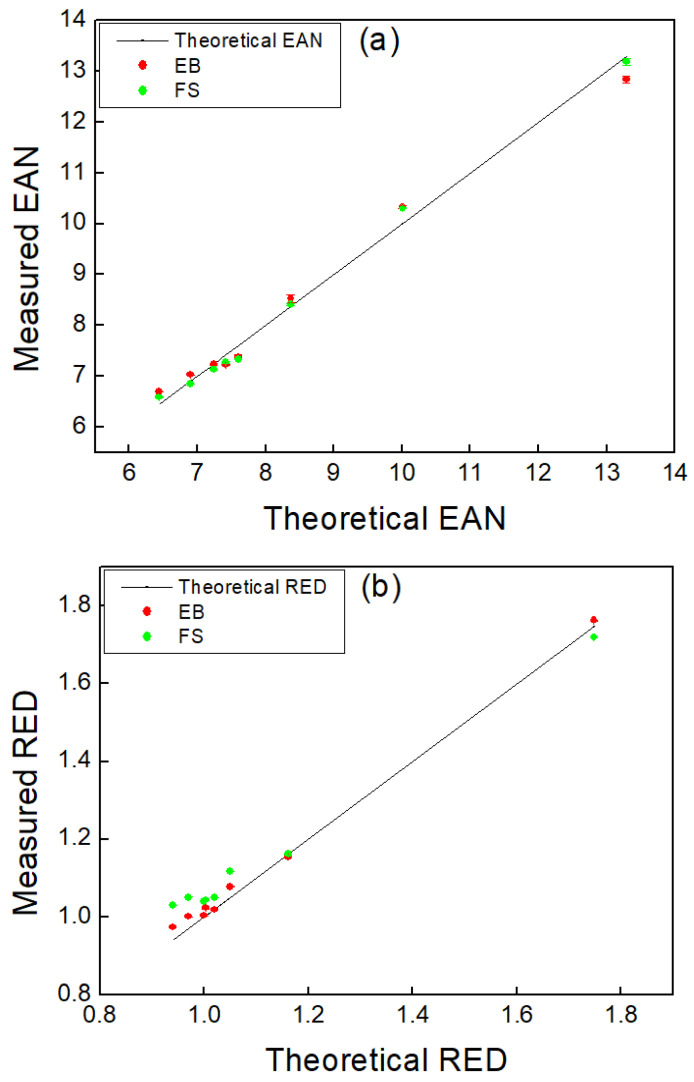
(**a**) Measured effective atomic number (EAN) and (**b**) relative electron density (RED) for energy bin (EB) and full spectrum (FS) modes. Mean errors of EAN for EB and FS are 2.48% and 2.04%, respectively. Mean errors of RED for EB and FS are 1.68% and 4.58%, respectively.

**Table 1 sensors-22-09220-t001:** Eight materials for stoichiometric calibration, the type of material, relative electron density, and effective atomic number are described.

No.	Materials	EAN ^1^	RED ^2^
1	Adipose	6.44	0.940
2	Breast	6.90	0.970
3	Solid Water	7.24	1.000
4	Brain	7.42	1.020
5	Liver	7.60	1.050
6	Iodine 2 mg/mL	8.37	1.003
7	Inner Bone	10.01	1.161
8	Cortical Bone	13.29	1.748

^1^ EAN: effective atomic number, ^2^ RED: relative electron density.

**Table 2 sensors-22-09220-t002:** Spectral parameters for extraction of effective atomic number (EAN) by stoichiometric calibration of dual-energy image in energy bin (EB) and full spectrum (FS) mode.

Material	EB ^1^	FS ^2^
k_1,L_	−0.0309	−0.0367
k_2,L_	0.0064	0.0049
k_1,H_	1.7090	1.4060
k_2,H_	0.1889	0.1065

^1^ EB: energy bin, ^2^ FS: full spectrum.

**Table 3 sensors-22-09220-t003:** Spectral parameter for extraction relative electron density (RED) of dual-energy image in energy bin (EB) and full spectrum (FS) mode.

Material	EB	FS
k_1_(E)	0.0107	0.0297
k_2_(E)	−1.538 × 10^−5^	−8.109 × 10^−5^

**Table 4 sensors-22-09220-t004:** Theoretical and experimental values for effective atomic number (EAN) and relative electron density (RED) for eight tissue-equivalent materials. Relative error values are indicated in parentheses.

No.	Materials	EAN			RED		
		Theory	EB	FS	Theory	EB	FS
1	Adipose	6.44	6.69 (3.95%)	6.60 (2.54%)	0.940	0.974 (3.57%)	1.030 (9.55%)
2	Breast	6.90	7.03 (1.91%)	6.84 (−0.82%)	0.970	1.001 (3.24%)	1.050 (8.27%)
3	Solid Water	7.24	7.24 (0.01%)	7.13 (−1.47%)	1.000	1.004 (0.38%)	1.040 (3.98%)
4	Brain	7.42	7.23 (−2.60%)	7.27 (−1.98%)	1.020	1.018 (−0.18%)	1.049 (2.82%)
5	Liver	7.60	7.38 (−2.89%)	7.33 (−3.49%)	1.050	1.077 (2.59%)	1.116 (6.29%)
6	Iodine 2 mg/mL	8.37	8.53 (1.86%)	8.42 (0.54%)	1.003	1.023 (2.03%)	1.044 (4.05%)
7	Inner Bone	10.01	10.33 (3.19%)	10.30 (2.93%)	1.161	1.153 (−0.65%)	1.161 (0.03%)
8	Cortical Bone	13.29	12.84 (−3.42%)	12.95 (−2.54%)	1.748	1.762 (0.81%)	1.719 (−1.64%)

## Data Availability

Not applicable.

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
