# Peer review of "Improving the Accuracy of the Effective Atomic Number (EAN) and Relative Electron Density (RED) with Stoichiometric Calibration on PCD-CT Images"

_sensors, 2022, doi:10.3390/s22239220_

Round 1
Reviewer 1 Report
This manuscript introduced an approach to improve the EAN and RED accuracy of PCD in CT by using PCD CT-based stoichiometric calibration. In my point of view, this manuscript is well organized with clear expression. This paper can be published on the journal, sensor. However, I’m not a specialist in the field of CT detection. The overall decision should also consider others’ opinions.
I have only one question about the result.
You just compare the result from EAN and RED (EB and FS) in figure 5 and 6. I can’t see the improvement of your algorithm (stoichiometric calibration).
Author Response
Response 1: Thank you for your valuable comments. We responded point by point to the comments from the reviewers. In addition, all modified contents are marked in red in the manuscript.
As you commented, we modified Figure 5 and 6 to show the improved results of EAN and RED obtained from stoichiometric calibration. Figure 5 shows the EAN and RED images in color, emphasizing the density of images in EB and FS mode. Figure 6 also highlights how close the experimental values of EAN and RED are to the true values.

Reviewer 2 Report
Kihong et al. reported the improving the accuracy of the effective atomic number and relative electron density with stoichiometric calibration on PCD-CT Images. The work is interesting.
But some presentations is overstated.
“According to the stoichiometric calibration used for curve fitting, EB had a R2 of 0.9970, which resulted in a higher accuracy than FS, which had a R2 of 0.9966.” The 0.9970 is very close to the value of 0.9966 if it is considered the system error, as your word in abstract that “The errors of both 17 EAN and RED obtained with EB were within 4%.”. “EB also produced a better result in curve fitting than FS, resulting in a R2 of 0.9722 versus 0.9636 for FS.”, which is more convincing.
Author Response
Response 1: Thank you for your valuable comments. We responded point by point to the comments from the reviewers. In addition, all modified contents are marked in red in the manuscript.
In the results of this study, the R2 of EB and FS obtained from stoichiometric calibration are similar. Therefore, the existing sentence was modified as follows.
(line 257 – 258 in Discussion section)
“According to the stoichiometric calibration used for curve fitting, EB had a R2 of 0.9970, and FS had a R2 of 0.9966.“

Reviewer 3 Report
Some remarks:
The study is quite interesting and suitable for publication in "sensors" with the application of photodetectors like CdTe.
I am wondering if one could give a clear explanation of the errors mentioned. The errors in figs. 4 and 6 are within the size of the points?
I propose to write more detailed about the Importance of the studies - like merits for dose calculation in particle beam therapy. One sentence in the Abstract and at the end of "conclusions" is not sufficient.
Author Response
Point 1: Some remarks:
The study is quite interesting and suitable for publication in "sensors" with the application of photodetectors like CdTe.
I am wondering if one could give a clear explanation of the errors mentioned. The errors in figs. 4 and 6 are within the size of the points?
Response 1: Thank you for your valuable comments. We responded point by point to the comments from the reviewers. In addition, all modified contents are marked in red in the manuscript.
As you commented, the errors in figs. 4 and 6 are within the size of the point. We modified the graphs in Fig. 6 to highlight how close EAN and RED obtained with EB and FS modes are to their theoretical values. We also slightly adjusted the sentences to match the modified graph in Figure 6. The adjusted sentences are as follows.
(line 219 – 228 in Results section)
Figure 6(a) show EAN obtained in the EB and FS mode. In EB mode, comparing the measured EAN with the theoretical value, the deviation was between -3.42% and 3.95%. The average error was 2.48% and the standard deviation was 1.23%. In FS mode, the measured EAN deviated from the range of -3.49% to 2.93% compared to the theoretical value. The average error was 2.04% and the standard deviation was 1.03%. As shown in Figure 6(b), RED were calculated using the EB and FS method. In EB mode, as compared to the theoretical value, the measured RED varied between -0.65% and 3.57%. The average error was 1.68% and the standard deviation was 1.35%. In FS mode, the measured RED varied from -1.64% to 9.55% compared to the theoretical value. The average error was 4.58% and the standard deviation was 3.26%.
Point 2: I propose to write more detailed about the Importance of the studies - like merits for dose calculation in particle beam therapy. One sentence in the Abstract and at the end of "conclusions" is not sufficient.
Response 2: Thank you for your valuable comments. In the Discussion section, we have added an answer to your comments as shown below.
(line 303 – 311 in Discussion section)
Unlike SECT, DECT is of interest as an imaging modality for proton therapy planning because it can discriminate changes in patient density and chemical composition. In the proton treatment, the stopping power ratio(SPR) of a given medium to water is used to calculate the water-euiqvalent path length (WEPL) of the patient’s tissue. WEPL is also used to calculate initial proton energies and adjust water equivalent dose to tissue dose. Therefore, the accuracy of SPR is important in contributing to the dose accuracy of protons. In order to improve the accuracy of SPR calculated from EAN and RED, a PCD based detector would be used to study the accuracy improvement of EAN and RED.
